# Development and External Validity of a Short-Form Version of the INICIARE Scale to Classify Nursing Care Dependency Level in Acute Hospitals

**DOI:** 10.3390/ijerph17228511

**Published:** 2020-11-17

**Authors:** Ana María Porcel-Gálvez, Sergio Barrientos-Trigo, Elena Fernández-García, Regina Allande-Cussó, María Dolores Quiñoz-Gallardo, José Miguel Morales-Asencio

**Affiliations:** 1Faculty of Nursing, Physiotherapy and Podiatry, University of Seville, 41009 Seville, Spain; aporcel@us.es (A.M.P.-G.); efernandez23@us.es (E.F.-G.); rallande@us.es (R.A.-C.); 2Research Group under the Andalusian Research, Development and Innovation Scheme CTS-1019 Complex Care, Chronic and Health Outcomes, Instituto de Biomedicina de Sevilla (IBIS), 41013 Seville, Spain; 3Emergency Unit, Hospital Universitario Virgen del Rocío, 41013 Seville, Spain; 4Instituto de Investigación Biosanitaria de Granada (IBS), Hospital Universitario Virgen de las Nieves, 18012 Granada, Spain; lolaquinoz@gmail.com; 5Faculty of Health Sciences, Instituto de Investigación Biomédica de Málaga (IBIMA), Universidad de Málaga, 29010 Málaga, Spain; jmmasen@uma.es

**Keywords:** assessment tools, dependency care, health measurements, hospitals, instrument development, nursing care, outcome assessment, psychometrics, scales, validation studies

## Abstract

*Background:* The increasing dependence care in patients hospitalized in acute hospitals around the world entails classification systems heeding the wide range of care dependency levels generated by the many different types of dependent patients. This article is a report of a study assessing the validity and reliability of a short-from version of the instrument (Inventario del NIvel de Cuidados mediante Indicadores de Resultados de Enfermería (INICIARE)) used to classify inpatients according to their care dependency level. *Methods:* The validation, carried out in a multicenter longitudinal study, included three different samples: the first sample of 1800 patients to evaluate the reliability and validity, a second of 762 patients for confirmatory factor analysis, and a third of 762 to test the short-form version. Patients over 16 years of age, admitted to medical or surgical units at 11 public hospitals, were included. *Results:* The final sample included 3605 patients. Patients had a mean age of 64.5 years, 60% were admitted to medical units, with severe dependency. The validation process yielded two versions of the instrument: a 40-item version, with eight factors with 83.6% of total variance explained and Cronbach’s alpha values between 0.98 and 0.92, and a short-form with 26 items, with five factors and Cronbach’s alpha values between 0.96 and 0.90. The Confirmatory Factor Analysis yielded a good fit model to the 40-item version (Chi Square on Degrees of Freedom CMIN/DF) = 1.335; Normed Fit Index (NFI); Tucker–Lewis Index (TLI); Comparative Fit Index (CFI) > 0.90; Standardized Residual Root Mean Square (RMSEA) = 0.02; and Standardized Residual Root Mean Square (SRMR) = 0.027) and 26-item version (Chi Square on Degrees of Freedom CMIN/DF = 1.385; NFI = 0.998; CFI = 0.999; RMSEA = 0.02; and SRMR 0.02). Both INICIARE versions obtained a high correlation between them (r = 0.96; *p* < 0.001). *Conclusion:* INICIARE has proved to be a valid and reliable instrument for the assessment of the level of care dependency of acutely hospitalized patients.

## 1. Introduction

Life expectancy has significantly increased over the last few decades. The World Health Organization (WHO) predicts that in 2050 people aged 60 or over will account for 33% of the world population, thus quadrupling the number of octogenarians and nonagenarians [1].

In addition, morbidity rates also increase with age, and this situation is associated with the current aging characteristics, such as the presence of chronic diseases, related to higher levels of dependency in elderly people [2]. The dependence can be defined as the loss of physical, psychological and functional capacity, and this situation increases the complexity of the care process and the care demand [3]. Increasingly health institutions are implementing the person-centered care framework, focusing on the person’s caring needs, in order to provide quality care and to improve the efficiency and effectiveness of health systems. This caring framework allows personalized care, considering the specific situation of the elderly people and their dependency caring needs [4].

Ageing and chronicity are continually challenging healthcare systems [5]. This leads not only to an increase in dependence care in hospitalized patients, but also to a complex caring process in acute hospitals around the world [6]. Regarding the provision of nursing care, this scenario calls for classification systems that take into account the wide range of care dependency level generated by the many different types of dependent patients [7]. Nursing care dependency may be defined as the care needs that patients have during their hospital stay. In turn, patient care needs result in nursing tasks that increase the workload of nurses [8]. 

Since the 1970s, numerous instruments have been designed to quantify the nursing workload. Most of these instruments, based on activities developed by the nursing staff, are assessed in time units, and use either direct or indirect methods [9]. Although cross-cultural validation studies of instruments from past times have proliferated, they continue to limit themselves to assessing certain components of the nursing practice in order to determine the nursing workload. These instruments could be classified into three large groups: (1) instruments based on the quality of care and safety, (2) instruments that are based on professional nursing competencies in hospitalization, and (3) instruments focusing on nursing care or care dependency [10].

Among them, the Care Dependency Scale (CDS), conceptually based on the level of patient dependency according to caring needs, deserves a special mention since it has been validated in the Spanish population [11] and in other countries such as Italy and France [12,13]. However, the CDS has been tested in very specific groups of patients, generally in the institutionalized elderly, and to a lesser extent in the hospitalized elderly suffering dementia. This limits its reliability in broader hospital settings focused on acute care and with more diverse patient typologies [11].

In this context, our research group developed an instrument to assess the care dependency level in acutely hospitalized patients. The Inventario del NIvel de Cuidados mediante Indicadores de Resultados de Enfermería (INICIARE) (level nursing care inventory through indicators of classification of Nursing Outcomes) was created to assess the physiological, instrumental and psychosocial dependency of hospitalized patients, from nurses’ clinical judgment, supported by a system of standardized language such as the Nursing Outcomes Classification [14,15].

Referring to the instrument’s classification aforementioned [10], INICIARE Scale could be considered as a new group of instruments based on a novel approach using the standardized nursing language (SNL) systems to identify different patient dependency statuses, namely the Nursing Outcomes Classification (NOC). This SNL sets out a framework to develop outcomes sensitive to nursing interventions [15]. The NOC is sensitive to changes in the patient’s condition and facilitates the assessment and documentation of health outcomes [15,16]. In addition, it has a codified and taxonomic structure that has facilitated its integration into the health services information systems and the creation of Nursing Minimum Data Sets [17,18]. The NOC also overcomes the usual limitations of classification models that do not share the same definitions [18,19].

One of the greatest advantages of this type of instruments is that is based on patients’ conditions and not on activities or tasks, therefore avoids institutional bias that usually are inherent to traditional nursing work time scales. Therefore, INICIARE scale overcomes several limitations which other instruments based on nursing procedures have, namely, those related to time dedicated to nursing tasks, and its application to acute patients, whether medical or surgical. Nevertheless, INICIARE scale was validated in a previous study where only considered methods of exploratory factor analysis, assessed in only one hospital, with the consequent external validity limitations [14].

Therefore, with the purpose of advancing the availability of instruments evaluated with solid methods and with adequate external validity, a short-from version of the instrument scale (INICIARE) to classify inpatients according to their care dependency level is used in this study.

## 2. Methods

### 2.1. Study Design and Participants

A longitudinal multicenter study was conducted between 2016 and 2018 to analyze the reliability and psychometric validity of the INICIARE scale. Firstly, hospitals were recruited based on their intention to participate in the study and patients were subsequently selected.

Hospitals: Ten public hospitals in Andalusian Healthcare System (southern Spain) participated in the study. The Andalusian Healthcare System, belonging to the Spanish National Health System, with 26 hospitals, provides universal and free coverage to 8.4 million inhabitants in the region. In Spain, there are hospitals that have different levels according to their specialization and reference population, and are classified into three types [20,21,22]. 

Hospital size: Three types of hospitals were selected: primary, specialties and tertiary. Primary hospitals (>500 beds and large metropolitan areas) are the highest-ranking hospitals and serve the entire population of the autonomous community and offer all clinical specialties. Specialties hospitals (between 200 and 500 beds and small metropolitan areas), cover the province in which they are located. They have a greater number of specialties than a tertiary hospital. Finally, tertiary hospitals (<200 beds and rural areas), which provide services to neighboring population centers, at most one hour away. 

Participants: The inclusion criteria were as follows, patients older than 16 years of age, both sexes, admitted to a medical or surgical hospitalization unit, and with a foreseen length of stay of over 48 h, to guarantee a minimal longitudinal follow-up. The time interval was long enough to prevent recall bias and short enough to ensure that patients had not changed with respect to the parameter to be measured. The following patients were excluded: women who required obstetric care (because of the specificity of their care needs), patients admitted to psychiatric services and critical care units.

### 2.2. Sampling and Sample

The sampling was carried out in a stratified, consecutive manner by (1) hospital size and (2) type of unit. Ten hospitals were finally included (3 Primary, 3 Specialties and 4 Tertiary) and the unit types were classified into Medical (*n* = 25) and Surgical (*n* = 25) units. We needed three different samples according to the validation process: sample (1) Exploratory Factor Analysis, sample (2) Confirmatory Factor Analysis and sample (3) Short-form version.

#### 2.2.1. Sample 1

According to the recommendations made by Iacobucci and Duhachek [23], the sample size must be related to the length of the scale. This sample was recruited in the year 2016. Regarding the exploratory factor analysis, according to Rouquette and Falissard [24], it is necessary to sample between 5 and 10 subjects per item. The scale of the initial version included 60 items, so between 300 and 600 subjects were needed. We decided to include 1800 patients for this exploratory phase.

#### 2.2.2. Sample 2

After the exploratory phase, we calculated the sample size needed for validating the factor structure by means of confirmatory factor analysis. This sample was recruited in year 2017. We estimated a sample size to contrast the null hypothesis H0: R = 0. Following the recommendations from MacCallum, Browne, and Sugawara [25], where R is the Standardized Residual Root Mean Square (RMSEA) of the independence model and R0 the observed RMSEA. Thus, with a power of 85%, an alpha value of 0.05, and 488 degrees of freedom being R = 0.05 and R0 = 0.045, 762 patients were necessary. 

#### 2.2.3. Sample 3

In 2018 an additional third sample of 762 patients was collected, in order to explore the validity and reliability of a short-form version of the resulting instrument for the confirmatory factor analysis (CFA) analysis.

All the three samples would be increased up to 5% to replace potential losses.

### 2.3. Data Collection Procedures

A study protocol was carried and the following three phases were established:

#### 2.3.1. Recruitment

The principal investigator along with the nursing directors of each hospital, selected 31 nurses trained to collaborate with the project (coordinating team). The coordinating team selected 157 registered nurses who participated voluntarily as data collectors, the number per hospital depending on hospital size (from 3–4 nurses in tertiary hospitals to 30–35 in primary hospitals). A total of 68 nurses from primary hospitals, 65 nurses from specialties hospitals and 24 nurses from tertiary hospitals. Nurses’ participation was voluntary, and they did not receive any compensation. 

#### 2.3.2. Training

The research leader team, with the support of at least one collaborating researcher from each hospital, conducted at least one training workshop at each hospital, two weeks prior to data collection. In the workshops, the nurses were presented with the background of the study (study objective, purpose and use of Nursing Outcome Classifications), the data collection procedure (clinical interview techniques, inclusion criteria, schedule, use of consent informed) and training in the administration of the different scales.

#### 2.3.3. Registration

During the sample collection period, all the patients’ data from the study unit were collected. The coordinating team were in charge of recording the data in the encrypted web platform designed for the study. This team was in constant contact with the registered nurses. Monthly reviews of the implementation of the registry and compliance with the schedule were established. This allowed evaluating and redirecting of the planned actions, minimizing biases and limitations. During this process, a copy of the protocol was always available in the participating Units for possible consultations.

### 2.4. Study Variables and Measures

The study variables were derived from:

#### 2.4.1. Participant Characteristics

Age, gender, functional and cognitive status: autonomy to perform daily activities (Barthel Index), cognitive impairment (Pfeiffer test) and length of stay.

#### 2.4.2. Hospital Characteristics

Hospital size (first primary hospitals, hospital specialties, tertiary hospitals), admission unit (medical or surgical).

#### 2.4.3. Instruments: INICIARE Scale and Care Dependency Scale (CDS)

The original version of INICIARE was comprised of 60 items and used a 5-point Likert scale. Score 1 reflects the least favorable situation, while score 5 reflects the most favorable. Each of the component variables is in a score range of 1 to 5. The lowest score on the scale is 60 points and the highest is 300. It is a hetero-administered scale. In the previous study, INICIARE obtained high inter-observer reliability ratings, with values between 0.80 and 0.99, a Cronbach’s alpha score of 0.92 (Intraclass Correlation Coefficient 0.830–0.964), and 74.12% explanatory power of the variance [26].

The CDS consists of 15 items that measure the level of patient dependency according to 15 concepts related to human needs using a 5-point Likert scale. CDS scores can range from 15 to 75 points. A score equal to or less than 68 points indicates dependent care, while a score greater than 68 points indicates independent care [12]. The psychometric validation demonstrated high internal consistency (Cronbach’s alpha = 0.97) as well as acceptable values of inter-observer (Kappa = 0.4–0.64) and intra-observer (Kappa = 0.55–0.8) reliability. The exploratory factor analysis (EFA) indicated the grouping of the items into a single factor with an explanatory power of the variance of 69.5% [11].

### 2.5. Ethical Approvals

Ethical aspects present in current legislation have been taken into account for this study. The ethics committee of the Andalusian Healthcare System (Code: 1967) approved the project. Patients were informed verbally and in writing of the objective and purpose of the study. Participants signed a declaration of informed consent and were aware of their right to withdraw from the study at any time. The anonymity of the patients was preserved at all times by using code numbers.

### 2.6. Data Analysis

All analyses were performed using the statistical packages: SPSS 25©, for statistical analysis and the exploratory factorial analysis (IBM Corp., Armonk, NY, USA) [27], AMOS 23©, to run the confirmatory factor analysis (IBM SPSS, Chicago, IL, USA) [28], and STATISTICA 12©, for complementary statistical analysis and graphic design (StatSoft., Tulsa, OK, USA) [29].

An exploratory analysis was carried out to evaluate the frequency, distribution, and normality of variables using the Kolmogorov–Smirnov test. A calculated value of 0.005, revealed that the scores were not normally distributed. 

#### Psychometric Testing

The psychometric analysis included reliability and validity tests (Figure 1). Internal consistency was calculated using Cronbach’s alpha coefficient and inter-observer reliability through the intra-class correlation coefficient (ICC). Inter-item and item-total correlation were also calculated. Items with poor inter-item and item-total correlations, or low communalities were removed from the model. Following this, an exploratory factor analysis (EFA) was developed, with principal axis factoring and both orthogonal and non-orthogonal rotations. Kaiser–Meyer–Olkin (KMO) and Bartlett’s sphericity test had already been performed to determine if the data were suitable for factor analysis. In addition, frequency of endorsement and possible occurrence of ceiling and floor effects were calculated.

With this exploratory solution, a confirmatory factor analysis (CFA) was carried out to evaluate the resulting factor structure hypothesis. An unweighted least square (ULS) estimation procedure was used because observed indicators did not follow a continuous and multivariate normal distribution [30]. In order to evaluate the goodness of fit of the confirmatory models, the following indices were used [31]: the penalizing function (Chi Square on Degrees of Freedom (CMIN/DF) (values ≤ 3 indicated a good fit); the RMSEA index (Root Mean Square Error of Approximation) (values ≤ 0.05 or 0.08 indicated a good fit); NFI (Normed Fit Index), CFI (Comparative Fit Index), TLI (Tucker–Lewis Index) (values ≥ 0.96 indicated a good fit), and SRMR index (Standardized Residual Root Mean Square) (values ≤ 0.80 indicated a good fit).

To establish the concurrent criterion validity, a Spearman’s rho test was conducted to determine the correlation between the INICIARE scores with the CDS scores and INICIARE scores and the Barthel Index. The starting hypothesis was that there is a strong, positive correlation between both scales. Additionally, a Bland–Altman plot was calculated with both measures, to determine their concordance. 

To establish the cut-off points, we used normative percentile scores for both versions which were calculated to establish normal values of care dependency for types of hospital. With a view to determining sensitivity to change, the differences between the INICIARE scores during stay and the different observations were analyzed by calculating Cohen’s d. 

## 3. Results

### 3.1. Participant Characteristics

The study sample consisted of 3605 patients, divided in three subsamples (*n* = 1901; *n* = 821; and *n* = 883, respectively) with similar features. Male gender was present among 53.8% and 57.4% in the three samples, with a mean age of 64.5 years, and a great majority admitted to medical units, with severe dependency according to the Barthel Index, and a good cognitive status according to the Pfeiffer test. The recruitment of the sample was opportunistic depending on the availability of the hospitals, units and nurses recruited who agreed to participate in the study. There were no losses, the response rate was 100%. The details of the three subsamples are described in Table 1.

### 3.2. Psychometric Testing

#### 3.2.1. Sample 1 (Initial 60-Item Scale to 46-Item Version Scale)

As show in Table 1, in sample 1, the mean of each of the items ranged between 3.26 and 4.57 (SD 0.85 to 1.60). No floor/ceiling effect was detected, since endorsement frequencies were below 75% in all cases.

The initial 60-item scale was reduced to a 46-item version after the analysis of inter-item and item-total correlations and their impact on Cronbach’s alpha, in addition to their communalities. An EFA was conducted on this last version, yielding a solution of eight factors with 83.6% of total variance explained. The KMO and Bartlett’s tests yielded appropriate values for EFA (KMO: 0.969 and *p* < 0.001, respectively) (Table 2).

#### 3.2.2. Sample 2 (46-Item Version Scale to 40-Item Scale)

Subsequently, a confirmatory factor analysis was performed on this eight-factor model, in a different sample of 821 patients, with a final version of 40 items (large form version). Six items were deleted after assessing the standardized covariance residuals (Figure 2).

The eight factors were: (1) Respiration (6 items), (2) Feeding (3 items), (3) Elimination (3 items), (4) Mobility (5 items), (5) Hygiene (3 items), (6) Sleep and rest (3 items), (7) Communication (4 items), and (8) Health behaviors (13 items). The fitness of this model yielded the following values: (Chi Square on Degrees of Freedom CMIN/DF = 1.335 (*p* = 0.22); NFI = 0.998; TLI = 0.999; CFI = 0.999; RMSEA= 0.02; and SRMR = 0.027. Cronbach’s alpha of each dimension was: Respiration = 0.96; Feeding = 0.92; Elimination = 0.93; Mobility = 0.97; Hygiene = 0.93; Sleep and Rest = 0.98; Communication = 0.98; and Health behaviors =0.98. 

#### 3.2.3. Sample 3 (40-Item Version Scale to 26-Item Scale)

Subsequently, a shorter version of the 26-item version (short-form version) was generated after having deleted those items that offered higher residuals in the covariance matrix (Figure 3). Inter-observer reliability showed an intraclass correlation coefficient of 0.96.

The factors isolated in the model were (1) Breathing (five items), (2) Feeding (six items), (3) Elimination (five items), (4) Mobility (six items), and (5) Perception and health behaviors (five items). This version obtained appropriate adjustment parameters: Chi Square on Degrees of Freedom CMIN/DF = 1.385; NFI = 0.998; CFI = 0.999; RMSEA = 0.02; and SRMR 0.02. Cronbach’s alpha coefficient of each dimension of the 26-item version was: Respiration = 0.96; Feeding = 0.93; Elimination= 0.90; Mobility = 0.94; and Health behaviors and communication = 0.94.

#### 3.2.4. Large form Version (40-Item Scale) vs. Short Form Version (26-Item Scale)

Both INICIARE versions, 40-items (large form) and 26-items (short form), obtained a high correlation between them: r = 0.96 (*p* < 0.001). The cut-off points are based on a percentage distribution and the Z scores for each variable of the instrument as a function of the level of dependence. The mean value of INICIARE-40 in the full sample was 162.6 (SD 35.3), and for INICIARE-26 106.9 (SD 21.6).

Therefore, patients with a high degree of dependence are found at the 25th percentile, while independent patients are at the 100th percentile. Considering these percentiles, the dependency intervals were established as follows according to the short or long version. Short version: High dependency 26–95, Moderate dependency 96–114, Risk of dependency 115–124 and Independence 125–130. Long version: High dependency 40–145, Moderate dependency 146–174, Risk of dependency 175–190 and Independence 191–200.

Normative percentile scores for both versions were calculated and are illustrated in Figure 4.

The scale INICIARE is a 40-item Likert scale, based on indicators of the NOC. For each indicator the score can range from zero to five: five reflects the most desirable patient’s condition, and one reflects the least desirable (Figure 5). The total score ranges from 40 points (indicating the highest level of dependence), up to 200 points, indicating independence. INICIARE-40 can be used at any time to assess the patient status. Nevertheless, the best option is to perform a measurement on patient’s admission and subsequent daily assessments. The score is set by the nurse according to his/her clinical judgment, based on the assessment data.

The scale INICIARE-26 is a 26-item Likert scale, based on indicators of the NOC. For each indicator the score can range from zero to five: five reflects the most desirable patient’s condition, and one reflects the least desirable (Figure 6). The total score ranges from 26 points (indicating the highest level of dependence), up to 130 points, indicating independence.

#### 3.2.5. Concurrent Criterion Validity

Regarding the concurrent criterion validity, INICIARE-40 and CDS obtained a Spearman coefficient of 0.87 (*p* < 0.001), and INICIARE-26 and CDS a coefficient of 0.88 (*p* < 0.001). The Bland–Altman analysis yielded a good concordance, although slightly less accurate in lower scores (Figure 7). 

When the correlation between the Barthel index (BI) and INICIARE-40 is analyzed, a strong and positive value with the BI is obtained (rho = 0.77), which indicates that both scales measure similar concepts (Figure 8). However, when the BI only correlates with the instrumental dimension of INICIARE-40, the value increases up to 0.802, which indicates a greater conceptual similarity (Figure 9).

The sensitivity to change of the instrument showed a Cohen’s d = 1.63 (*p* < 0.001; r = 0.63). INICIARE encompasses methodological strengths such as sensitivity to detect changes in patient status during hospitalization, a robust process of validation, and excellent psychometric properties evidenced after administering the (COnsensus-based Standards for the selection of health status Measurement INstruments (COSMIN) checklist [32], with the following results: A (Content validity) = +++; B (Structural validity) = +++; C (Internal consistency) = +++; D (Cross-cultural validity) = Not applicable; E (Reliability) = +++; F (Criterion validity) = +++; G (Measurement error) = +++; H (Hypothesis testing for construct validity)=+++, I (Responsiveness) = Not applicable.

Regarding discriminant validity, INICIARE (40-items and 26-item versions) was able to detect different levels of dependence among older patients. Thus, patients under 65 years obtained mean INICIARE-26 scores 114.3 (SD 17.8), versus 101.4 (SD 22.6) in those patients over 65 years (mean difference 12.91; *p* < 0.001). Similarly, INICIARE-40 scores were significantly lower among patients over 65 years: 153.3 (SD 37.3), 175.2 (SD 27.9), (mean difference 21.9; *p* < 0.001), respectively.

## 4. Discussion

The purpose of this study was to test the external validity and reliability of the short-form version of an instrument (INICIARE) to classify hospitalized patients according to their care. This has allowed the development of a more parsimonious and feasibility tool, due to the reduction in its factors and items with respect to the original scale of 60 items [14].

According to the participant characteristics, the result of this study indicates that patients admitted to hospitals in medical and surgical units are older (64.5 years) and have levels of cognitive impairment and advanced dependence. These results coincide with the aging of the population and the increase in chronic pathologies that translate into hospitalization of patients with increasingly advanced ages and who need more nursing care [33].

Concerning the structure, the scale items were reduced to two versions: 40 items and 26 items. The extended 40-item version could be used when exhaustive assessment of the nursing patient dependence status is necessary, or in those organizations with clinical records that include NOC in their information system, so that automated processing could be easily carried out. The short-form version presents similar extension to CDS, which facilitates its usage by clinical nurses, and nurse managers. These two versions are a great improvement with respect to the original 60-item version in terms of usability, validity and reliability [14]. 

The findings in this study demonstrate that the INICIARE instrument and short-form versions (INICIARE-26 and INICIARE-40) are valid and reliable scales based on the nursing patient’s dependency level. These results are of significant relevance since a total sample of more than 3605 patients was used; few studies provide sample sizes greater than 200 subjectsas can be seen in Iacobucci and Duhachek [23].

Regarding the CDS scale, it was demonstrated to have excellent psychometric properties in its validation studies. However, the validation of the CDS in very specific groups of patients, such as institutionalized elderly with dementia, prevents its use in acute settings and in patients with different cognitive and clinical profiles [9] INICIARE-26 and INICIARE-40 have demonstrated its validity to evaluate levels of dependence of patients with different age profiles, clinical conditions, and admitted to hospitals with diverse types of units, medical or surgical. Therefore, this scale in both short-version forms is the best instrument to measure results sensitive to nursing practice based on the dependence of nursing care in hospitalized patients [34].

Moreover, the correlation between the INICIARE scale and the IB scale was strong (0.77), and it supports the hypothesis that INICIARE could be used to evaluate the level of dependence. In addition, because BI only measures instrumental dependence, the research group decided to perform the correlation by grouping those INICIARE factors that measure the same dependence aspect, obtaining a strong correlation (0.8). A similar method was followed with the Northwick Park Dependency Score (NPDS) scale, obtaining a very strong correlation (rho = 0.91). This fact could be interpreted because the Barthel scale was developed by physiotherapists and they oriented its use towards physical or instrumental dependence [35]. 

After 50 years of research in the development of such instruments, the use of an a priori hypothesis in the formulation of items based on nursing actions or activities, which are used as proxies of the patient status, is a limiting factor [36]. The results of this research indicate that both the INICIARE instrument and the short- form version are based on a conceptual approach that sought to avoid this limitation, due to the institutional bias generated by the variability of practice styles [34]. In our view, this has been the key factor that has undermined the progress on nursing research in this area. Furthermore, the use of normative percentile scores allows the adaptation of the scale in different contexts [37].

In addition, having normative percentile scores available at the hospital allows the cut-off points to be calculated and thus a proper classification of patients based on their level of dependency. It would be a direct way of adapting the scale to specific contexts [37,38]. Nurses could use this to deliver specialized care to highly dependent patients. Furthermore, it could be used for patients themselves to find percentile graphs as an important tool for self-care. In general, our study is useful to establish useful reference values for dependency care in Andalusian Healthcare System (southern Spain). Future studies in different systems and contexts could achieve greater precision regarding the reference values.

In short, INICIARE is a tool that measures caring dependence level, testing the practical use of Standardized Nursing Languages (SNLs) in the patients’ assessment, under a conceptual model. This structure provides an important step in linking models and their indicators, being the empirical demonstration of a theory in nursing practice. It is necessary to delve into research that combines theory and practice and that, from a nursing framework, develop intermediate theories and vice versa [39]. Already in 1975, Rosnay affirmed that the models were the beginning points, and it is necessary to adapt the theoretical frameworks to reality [40]. Concretely, in Phaneuf’s adaptation, they considered the dependence as a six-level continuum, from independence to total dependence [41]. Independence has only one level while dependency is graduated in five levels, analogous to the classification of dependency by INICIARE. Therefore, if we start from this premise and based on Rosnay’s assertion, INICIARE is the substrate to propose a new middle range theory. 

In addition, the use of Standardized Nursing Languages (SNLs) for the formulation of the INICIARE items could favor the reduction in arbitrariness in decision making, enable a more flexible adaptation to different settings, and facilitate compatibility with digital information systems [40,41,42,43,44,45]. One important finding in this study was the fact that the short-form version evaluates those clinical dimensions of the patient that nurses deal with on a regular basis (breathing, feeding, elimination, mobility, health behaviors, etc.) A high number of nursing diagnoses are related with these domains [46] and, as a consequence, numerous nursing care studies focus on them [47,48]. Thus, it becomes a patient-centered classification system based on their nursing care dependency, making it possible to adjust the distribution of nursing staff in hospitals according to patient nursing care dependency. Since this instrument is based on patient status and not on nursing activities, it is not subject to institutional bias due to variations in patterns of clinical practice and protocols. With regard to inter-observer reliability, there is good consistency in the individual and global items, as evidenced by the use of the NOC in studies that have evaluated this aspect [14,49,50,51].

Therefore, this scale could improve the way of delivering personalized care to each patient. Health care policy-makers should use this instrument for a patient’s assessments during their admission process, to know the patients care dependency status, in order to optimize human and material resources. Our research group is immersed in developing and validating the Nursing Iniciare_Patient model (NIP 3.0) for the assumption of nurse resources adjusted for dependency levels in inpatient care evaluated with the IINICIARE scale [52]. With this model we aim to achieve improvement of staff outcomes and reduction in adverse events in the health care system. In short, the use of this model could modify the organizational culture of hospitals.

### Limitations

INICIARE short-form versions (INICIARE-26 and INICIARE-40) were validated in hospital settings, in surgical and medical units, as well as in adult patients. Consequently, its applicability is unclear in other settings, such as primary healthcare, mental health, or in other types of patient, such as obstetric, critical, or pediatric care. Further studies should endeavor to evaluate INICIARE in these settings. Therefore, some limitations have been found in the methodology such as the training of nurses and the distribution of the sample between types of hospitals and their temporality. 

The instrument can be used both for the initial assessment, and for surveillance and follow-up of the patient status. However, it cannot replace clinical nursing judgment in patient care and surveillance. Moreover, INICIARE short- form version (INICIARE-26 and INICIARE-40) are based on the evaluations performed by nurses, but a mismatch could take place between this evaluation and the patients’ perception about their care needs. In addition, it would be necessary for future studies to compare the agreement between the need of care perceived by patients, and the dependence level reported by the assessment with short-form version scale. Furthermore, future studies should be undertaken to evaluate the relation between levels of dependence, nurse staffing and patient outcomes.

## 5. Conclusions

INICIARE has demonstrated good external validity in acute hospitals with different sizes, geographical areas (metropolitan and rural areas), and specialty levels (primary, specialties and tertiary) and a wide variety of units (medical and surgical). The short-form versions (INICIARE-26 and INICIARE-40) are a valid and reliable instrument for the assessment of the nursing care dependency level of acute hospitalized patients, based on patient status. The validation process yielded two versions of the instrument: a 40-item version and a short-form version with 26 items been demonstrated to be a valid and reliable instrument for the assessment of the level of care dependency of acutely hospitalized patients. Therefore, the INICIARE short-form version (INICIARE-26 and INICIARE-40) will facilitate the comparison of the distribution of patients of the same levels of nursing care dependency between different hospital units in relation to nursing resources allocation. 

## Figures and Tables

**Figure 1 ijerph-17-08511-f001:**
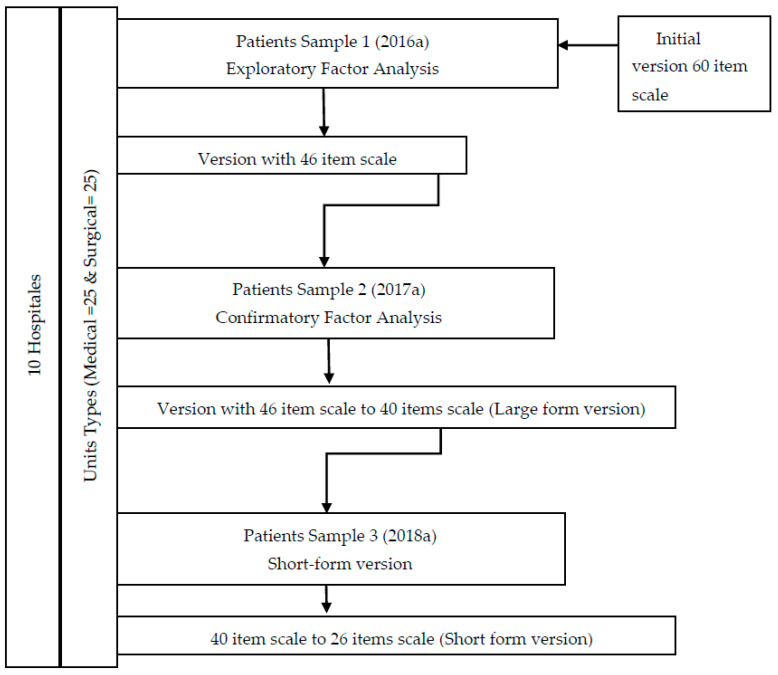
Description process two versions of the instrument: a 40-item and 26-item version.

**Figure 2 ijerph-17-08511-f002:**
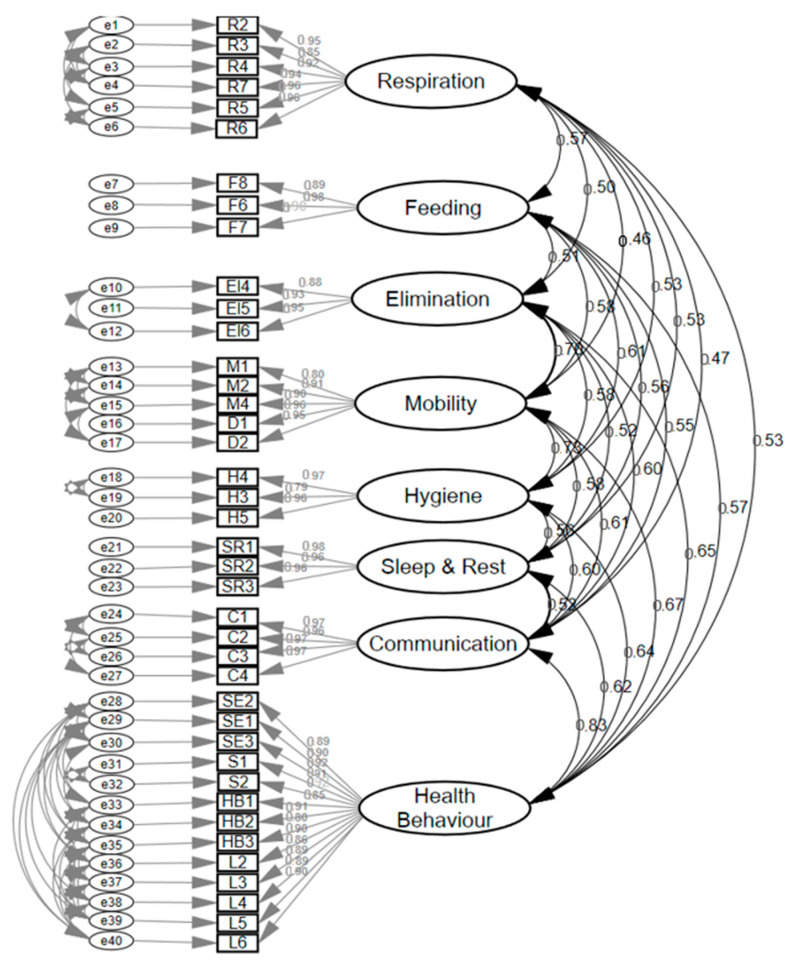
Confirmatory factor analysis 40-item scale (large form version).

**Figure 3 ijerph-17-08511-f003:**
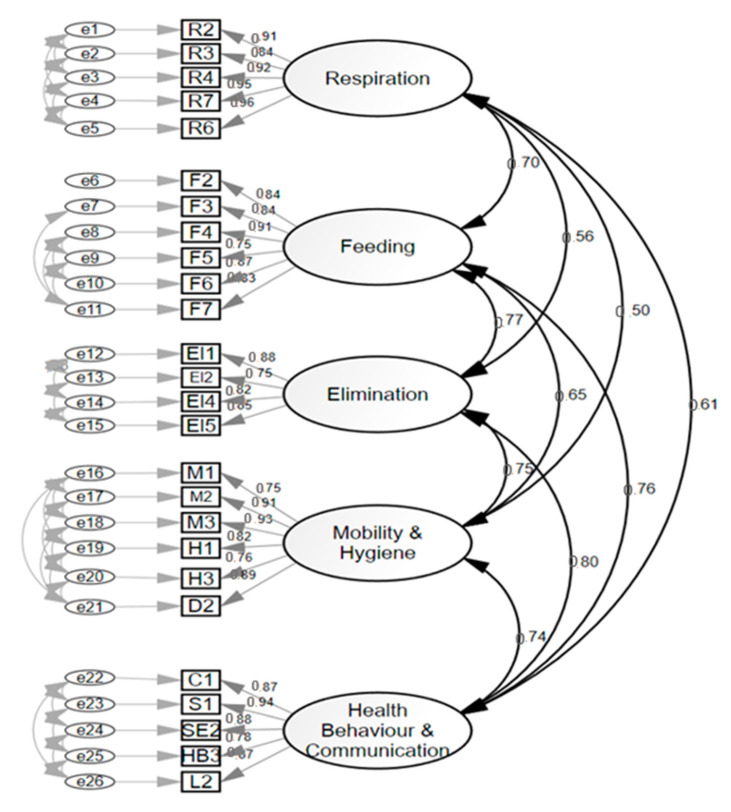
Confirmatory factor analysis 26-item scale (short-form version).

**Figure 4 ijerph-17-08511-f004:**
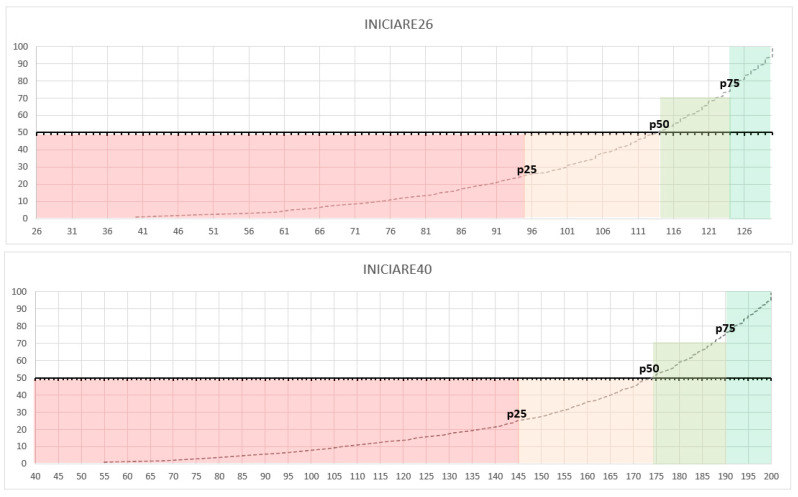
Percentile distribution Inventario del NIvel de Cuidados mediante Indicadores de Resultados de Enfermería (INICIARE)-26 and INICIARE-40.

**Figure 5 ijerph-17-08511-f005:**
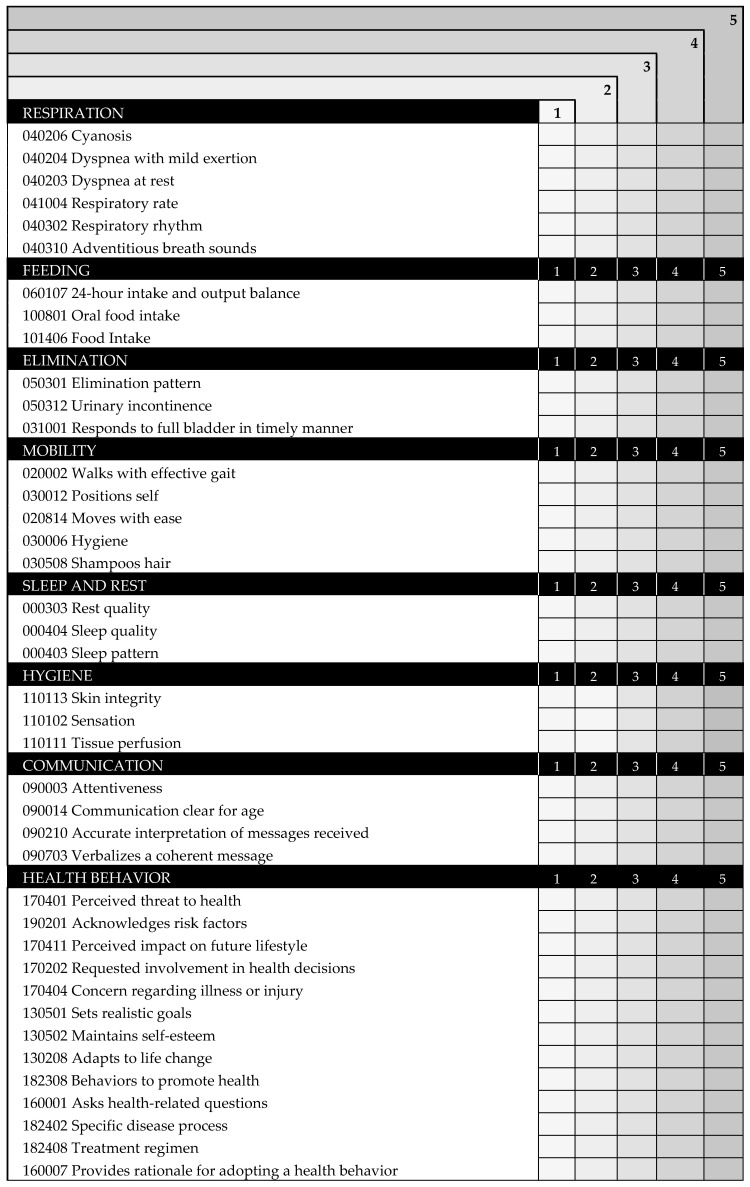
INICIARE-40 English Version.

**Figure 6 ijerph-17-08511-f006:**
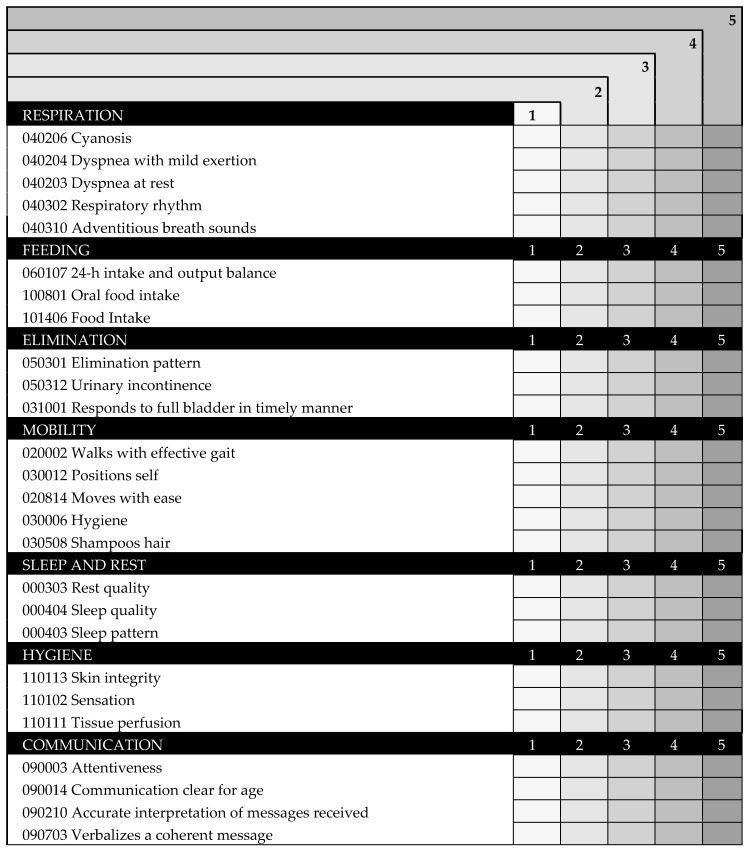
INICIARE-26 English Version.

**Figure 7 ijerph-17-08511-f007:**
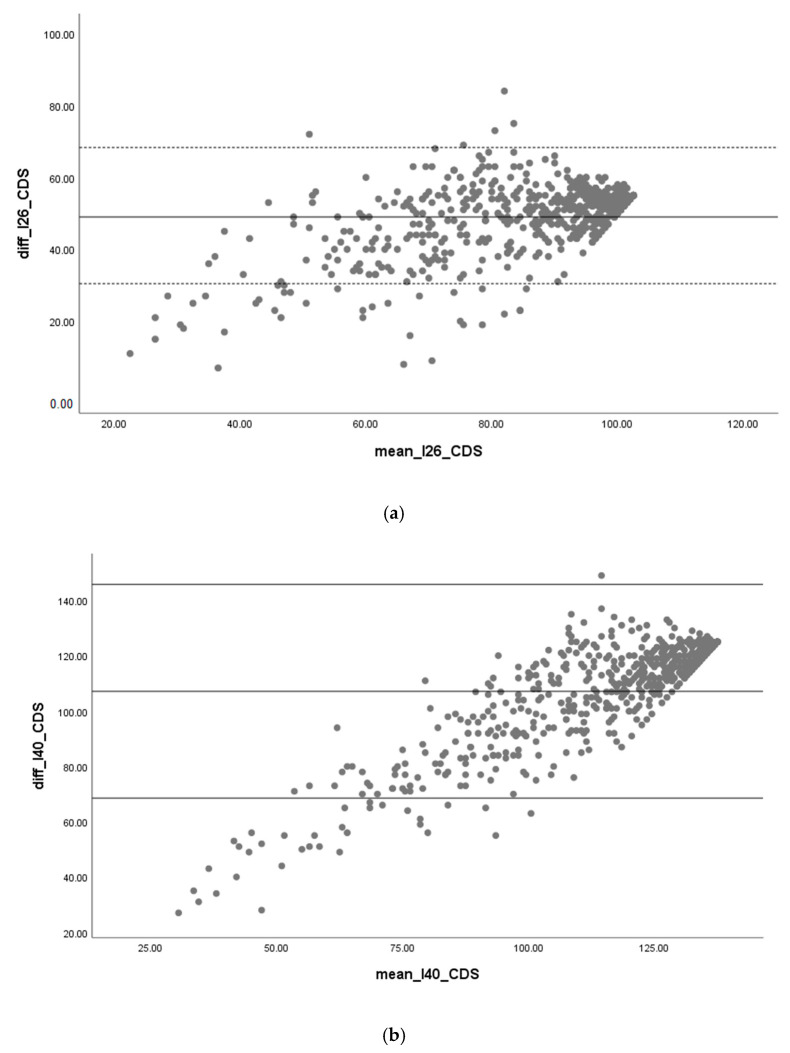
(**a**) Criterion validity INICIARE-26 and (**b**) INICIARE-40 with Care Dependency Scale (CDS).

**Figure 8 ijerph-17-08511-f008:**
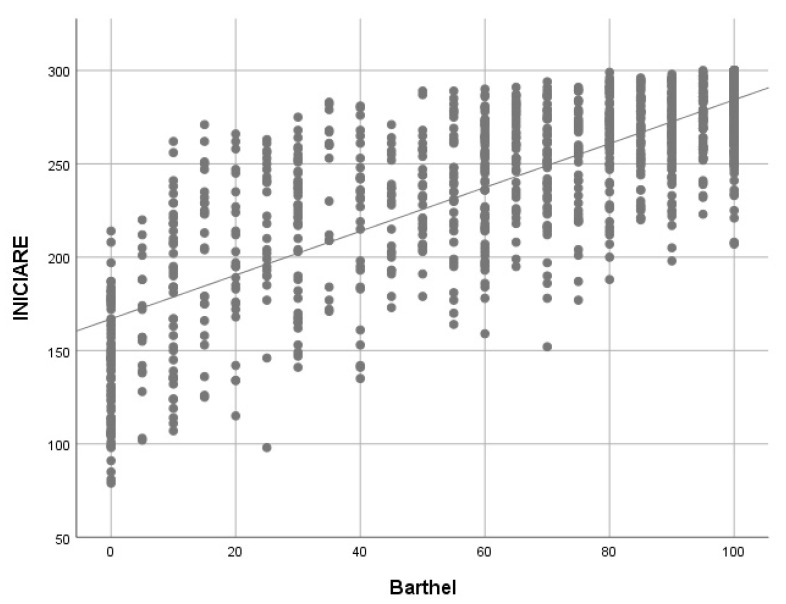
Barthel Index and INICIARE correlation with Spearman’s rho. rho = 0.763; *p* < 0.001.

**Figure 9 ijerph-17-08511-f009:**
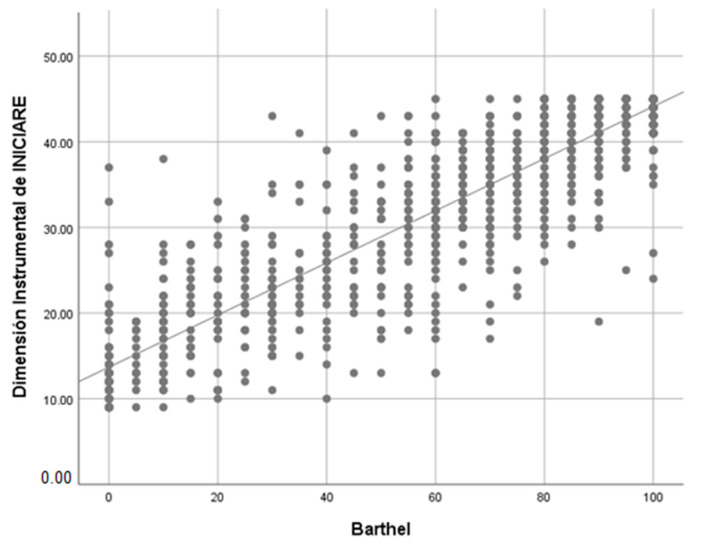
Barthel Index and Instrumental dimension INICIARE correlation with Spearman’s rho. rho = 0.826; *p* < 0.001.

**Table 1 ijerph-17-08511-t001:** Characteristics of the three subsamples.

Variables	Patients			Clinical Nurses
	Sample 1	Sample 2	Sample 3	Sample
	*n* = 1901	*n* = 821	*n* = 883	*n* = 157
	*n* (%) or Mean (SD)	*n* (%) or Mean (SD)	*n* (%) or Mean (SD)	*n* (%) or Mean (SD)
Age	64.7 (17.0)	64.5 (17.0)	64.5 (16.9)	
Gender				
Male	1022 (53.8)	471 (57.4)	489 (55.4)	
Female	879 (46.2)	350 (42.6)	394 (44.6)	
Unit				
Medical	1191 (62.7)	497 (60.5)	536 (60.7)	
Surgical	710 (37.3)	324 (39.5)	347 (39.3)	
Functional and cognitive status				
Barthel Index	54.3 (35.7)	58.5 (36.8)	55.9 (35.9)	
Pfeiffer	1.1 (0.3)	1.1 (0.3)	1.1 (0.3)	
Hospital				
Primary	845 (44.4)	404 (49.2)	412 (46.6)	68 (43.3)
Specialties	622 (32.7)	306 (37.3)	309 (34.9)	65 (41.4)
Tertiary	434 (22.8)	111 (13.5)	162 (18.3)	24 (15.3)

**Table 2 ijerph-17-08511-t002:** Structure matrix of the rotated solution 46-item version scale.

No.		Factor							
		1	2	3	4	5	6	7	8
1	160007 Provides rationale for adopting a health behaviour	0.880							
2	182402 Description of specific disease process	0.874							
3	160601 Claims decision-making responsibility	0.873							
4	182408 Description of treatment regimen	0.870							
5	182308 Behaviors to promote health	0.858							
6	160001 Asks health-related questions	0.852							
7	170202 Requested involvement in health decisions	0.823							
8	130501 Sets realistic goals	0.816							
9	130208 Adapts to life change	0.794							
10	170411 Perceived impact on future lifestyle	0.777							
11	130502 Maintains self-esteem	0.774							
12	170404 Concern regarding illness or injury	0.768							
13	190201 Acknowledges risk factors	0.731							
14	170401 Perceived threat to health	0.713							
15	040302 Respiratory rhythm		0.883						
16	041004 Respiratory rate		0.871						
17	040309 Accessory muscle use		0.857						
18	040310 Adventitious breath sounds		0.844						
19	040203 Dyspnea at rest		0.829						
20	040204 Dyspnea with mild exertion		0.794						
21	040206 Cyanosis		0.750						
22	030012 Positions self			0.816					
23	020814 Moves with ease			0.814					
24	020002 Walks with effective gait			0.804					
25	030002 Dressing			0.802					
26	030211 Removes clothes from upper body			0.793					
27	020802 Body positioning performance			0.723					
28	101406 Food Intake				0.857				
29	100801 Oral food intake				0.837				
30	101016 Food acceptance				0.678				
31	060107 24-h intake and output balance				0.549				
32	050301 Elimination pattern (urinary)					0.728			
33	050312 Urinary incontinence					0.688			
34	031001 Responds to full bladder in timely manner					0.684			
35	060211 Urine output					0.640			
36	050002 Maintains control of stool passage					0.415			
37	000404 Sleep quality						0.842		
38	000303 Rest quality						0.819		
39	000403 Sleep pattern						0.814		
40	110102 Sensation							0.753	
41	110111 Tissue perfusion							0.722	
42	110113 Skin integrity							0.640	
43	090703 Verbalizes a coherent message								0.655
44	090014 Communication clear for age								0.643
45	090003 Attentiveness								0.608
46	090210 Accurate interpretation of messages received								0.601

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
