# Peer review of "Development and External Validity of a Short-Form Version of the INICIARE Scale to Classify Nursing Care Dependency Level in Acute Hospitals"

_ijerph, 2020, doi:10.3390/ijerph17228511_

Round 1

Reviewer 1 Report

I appreciate the invitation to review this manuscript, and congratulate the authors for this well-designed and well-executed research. The study is very well structured, organized and written.

It is a topic of interest, and basing it on conceptual and methodological approaches to nursing is a significant contribution to knowledge.Stratifying hospitalized patients according to level of care is not a new approach, but it is important, and the design improves other previous studies by simplifying the assessment of patients and basing the evaluation of needs according to outcome criteria, assigning patients to a subgroup according to the level of care they will need.

Stratifying hospitalized patients according to level of care is not a new approach, but it is important, and the design improves other previous studies by simplifying the assessment of patients and basing the evaluation of needs according to outcome criteria, assigning patients to a subgroup according to the level of care they will need.

Abstract
If possible, I recommend structuring the abstract in natural sections.

Introduction
I recommend improving the relationship between the first and second paragraph of the introduction; they are currently presented separately and I don't think they are two different ideas. They could be better linked, or merged.
Given that the concept of dependent care is addressed, I think it would be convenient to theoretically frame the study, with a small more theoretical paragraph (a pure background) that would provide a foundation for dependent care.

Methodology: the design and development of the method section is impeccable. It is careful, structured, and describes perfectly how the study has been done. I believe, however, that it is excessively long, and that the authors should make an effort to simplify it a little and make it shorter.In that sense, you describe many characteristics that justify the reason for each decision of the method; but in many of those cases it could be dispensed, simplifying an unusually extensive method section.
Otherwise, technically it is excellent.

Results:
Could you incorporate the total response rate and for each subsample?
The rest of the results are presented properly, accurately, and in a logical order. The analyses are adequate, and allow results to be achieved that respond to the objectives of the study.

Discussion:
It is brief, concrete, and appropriate for a validation study.I would suggest, if you consider it convenient, to slightly extend the practical transference and applicability of the results of this study: in addition to its value for hospitals in particular, how can using tools for stratifying patients help the global health system? Will it focus on adapting care for each patient? Could or should it be used to modify the organization, the organizational culture of hospitals?

Author Response

Dear Editor in Chief and reviewers of the International Journal of Environmental Research and Public Health: Development and external validity of short-form version of the INICIARE scale to classify nursing care dependency level in acute hospitals.

Firstly, I would like to thank you for considering the manuscript for peer review. Similarly, I would like to thank you and the reviewers for their valuable contributions and comments made regarding the above-mentioned manuscript. All suggestions have been carefully considered and all modifications have been made as necessary.

Reviewer 1

Comments and Suggestions for Authors

I appreciate the invitation to review this manuscript, and congratulate the authors for this well-designed and well-executed research. The study is very well structured, organized and written.It is a topic of interest, and basing it on conceptual and methodological approaches to nursing is a significant contribution to knowledge. Stratifying hospitalized patients according to level of care is not a new approach, but it is important, and the design improves other previous studies by simplifying the assessment of patients and basing the evaluation of needs according to outcome criteria, assigning patients to a subgroup according to the level of care they will need.

Stratifying hospitalized patients according to level of care is not a new approach, but it is important, and the design improves other previous studies by simplifying the assessment of patients and basing the evaluation of needs according to outcome criteria, assigning patients to   subgroup according to the level of care they will need.

Abstract 
If possible, I recommend structuring the abstract in natural sections.

Introduction
I recommend improving the relationship between the first and second paragraph of the introduction; they are currently presented separately and I don't think they are two different ideas. They could be better linked, or merged.
Given that the concept of dependent care is addressed, I think it would be convenient to theoretically frame the study, with a small more theoretical paragraph (a pure background) that would provide a foundation for dependent care.

Methodology: the design and development of the method section is impeccable. It is careful, structured, and describes perfectly how the study has been done. I believe, however, that it is excessively long, and that the authors should make an effort to simplify it a little and make it shorter.In that sense, you describe many characteristics that justify the reason for each decision of the method; but in many of those cases it could be dispensed, simplifying an unusually extensive method section. 
Otherwise, technically it is excellent. 

Results: 
Could you incorporate the total response rate and for each subsample? 
The rest of the results are presented properly, accurately, and in a logical order. The analyses are adequate, and allow results to be achieved that respond to the objectives of the study.

Discussion: 
It is brief, concrete, and appropriate for a validation study.I would suggest, if you consider it convenient, to slightly extend the practical transference and applicability of the results of this study: in addition to its value for hospitals in particular, how can using tools for stratifying patients help the global health system? Will it focus on adapting care for each patient? Could or should it be used to modify the organization, the organizational culture of hospitals?

Authors: Thanks for all your comments. We really appreciate that you have found that the importance to assessment of patients according to outcome criteria and to the level of care.

The authors have considered all the recommendations and substantially improved the manuscript. Regarding to Abstract section have been structuring in natural sections. It has been a great opportunity to review the Introduction section, in order to improve the relationship between the first and second paragraph. We fully agree that it was necessary to achieve more in depth theoretical frame the study.

Regarding the methodology section, we wanted to be very thorough and explain in detail the process, we regret it has been too extensive. In results section It has been clarified a total response rate, there were no losses. We hope everything is clear now.

Finally, we agree with your opinion that the discussion parts of the manuscript must be extended and we included the explanation about the implication for the researchers and health care policy-makers. We hope that the modifications meet your expectations and that you can continue to consider the manuscript for publication in your journal.

Thank you for your time and consideration.

Sincerely, the authors.

Reviewer 2 Report

This article is very interesting for the development of nursing competencies to improve the quality of care.
The methodology of the work is adequate and its conclusions are also appropriate. However, some points and suggestions need to be made:
Methodology:
1. In the description of the hospitals participating in the study and the sample sizes, it is understood that the distribution of the sample is homogeneous between hospitals. However, there seems to be a bias in the distribution. It is not specified which sample corresponds to each type of hospital and its temporality.
2. In the recruitment of the participating nurses, the distribution by type of hospital, temporality of the sample, and how many patients were assigned to them is not specifically described. In this respect, why is the recruitment so uneven if the number of patients is apparently similar?
3- As for the training of nurses, no quality control mechanism is mentioned. How is a similar learning curve guaranteed among nurses? Furthermore, in figure 1 it is understood that it is a 3-year study.
4- About the instruments: According to the authors, the scale is heteroadministered, and the subsequent evaluation through COSMIN, a tool described to evaluate instruments whose assessor is the patient, may conflict with this. On the other hand, the NOC uses items that must be answered by patients with items that must be observed and judged by nurses.
5- It is not specified how the levels of patient dependency will be established from the data obtained. It is not specified how these cut-off points are to be analysed. For example, by means of a ROC curve.
Results:
1-Table 1 does not specify the sample losses, nor the distribution of patients by nurses and type of hospital.
2- When is a patient considered to be dependent? How are they classified by level? The supplementary tool "Appendix 2" should be part of the work and not presented as an appendix. It presents a range between 26 and 130 points. What are the cut-off points for considering a medium, high or low dependency. From which score is it necessary to make a clinical decision? Given that the NOC scales have variability in their interpretation, have the authors considered this variability in the construction of the instrument?
3-As the objective is to create an instrument for classifying patients (lines 82-83 and 281-283) and in fact it is specified that one of the variables is the Barthel index that is usually used for this purpose, are not different scores considered for the classification of dependency?
Limitations:
1- Some of the issues observed in the methodology such as the training of nurses and the distribution of the sample between types of hospitals and their temporality should be considered in the limitations of the study.

Author Response

Dear Editor in Chief and reviewers of the International Journal of Environmental Research and Public Health: Development and external validity of short-form version of the INICIARE scale to classify nursing care dependency level in acute hospitals.

Firstly, I would like to thank you for considering the manuscript for peer review. Similarly, I would like to thank you and the reviewers for their valuable contributions and comments made regarding the above-mentioned manuscript. All suggestions have been carefully considered and all modifications have been made as necessary.

Reviewer 2

Methodology: 
1. In the description of the hospitals participating in the study and the sample sizes, it is understood that the distribution of the sample is homogeneous between hospitals. However, there seems to be a bias in the distribution. It is not specified which sample corresponds to each type of hospital and its temporality. 
2. In the recruitment of the participating nurses, the distribution by type of hospital, temporality of the sample, and how many patients were assigned to them is not specifically described. In this respect, why is the recruitment so uneven if the number of patients is apparently similar?
3- As for the training of nurses, no quality control mechanism is mentioned. How is a similar learning curve guaranteed among nurses? Furthermore, in figure 1 it is understood that it is a 3-year study.
4- About the instruments: According to the authors, the scale is heteroadministered, and the subsequent evaluation through COSMIN, a tool described to evaluate instruments whose assessor is the patient, may conflict with this. On the other hand, the NOC uses items that must be answered by patients with items that must be observed and judged by nurses.
5- It is not specified how the levels of patient dependency will be established from the data obtained. It is not specified how these cut-off points are to be analysed. For example, by means of a ROC curve.

Results:
1-Table 1 does not specify the sample losses, nor the distribution of patients by nurses and type of hospital. 

2- When is a patient considered to be dependent? How are they classified by level? The supplementary tool "Appendix 2" should be part of the work and not presented as an appendix. It presents a range between 26 and 130 points. What are the cut-off points for considering a medium, high or low dependency. From which score is it necessary to make a clinical decision? Given that the NOC scales have variability in their interpretation, have the authors considered this variability in the construction of the instrument? 

3-As the objective is to create an instrument for classifying patients (lines 82-83 and 281-283) and in fact it is specified that one of the variables is the Barthel index that is usually used for this purpose, are not different scores considered for the classification of dependency?

Limitations:
1- Some of the issues observed in the methodology such as the training of nurses and the distribution of the sample between types of hospitals and their temporality should be considered in the limitations of the study.

Authors: Thanks for all your comments and suggestions. The authors have considered all the recommendations, reviewed the entire manuscript and substantially improved it.

Firstly, regarding to the methodology section we are really sorry that we made a mistake in the wording of simple calculation 1. We have deleted the statement. Thank you for this advice. Regarding to recruitment of the sample, was opportunistic depending on the availability of the hospitals, units and nurses recruited (we added this phrase in the results section). In table 1, we have added a further row included the distribution of three samples for each type of hospital (primary, specialties and tertiary).

Furthermore, in the recruitment of the participating nurses, in 2.3.1. Recruitment (methodology section) we have been included and explained this information. We hope the clarification of patient sampling and nurse’s participants clarify this section.

According to the training of nurses, we have extended the paragraph corresponding to 2.3.2 Training in methodology, we hope everything is clear now.

About the instruments, we consider that COSMIN checklist is the best method to evaluate the methodological quality of studies on measurement properties. We agree that the last version recommends the use in the patient-reported outcome measures, however numerous systematic reviews of validation studies use this method independently of instrument evaluated, for this reason we have included COSMIN checklist in this study.

We fully agree that is it was necessary to specified how the levels of patient dependency will be established from the data obtained, we have introduced in methodology in statistical analysis: To calculate the cutoff points, normative percentile scores for both versions have been calculated were used. This topic has also been discussed (line 368-line 376)

Regarding to the result section, we have included the distribution of patients by type of hospital and we have included the nurses participants. We hope the results are now better understood.

Furthermore, thank you for this comment about your question when patient considered to be dependent. Regarding this topic, since the level of dependence and the use of the NOC as a methodological basis are important points of the study. Indeed, the paragraph on results was short, as well as the discussion of these aspects. We have therefore introduced new paragraphs to clarify this content.

In methodology, in the statistical analysis section, the correlation INICIARE scores and the Barthel Index was performed. In results we have added the 283-294 line and included two new figures (figure 6, figure 7) and have included in discussion section.

Regarding the tool INICIARE (short and long version), we agree that it is the result of the research being the most important result but its extension led us to the decision to present them as an appendix. However, we agree to present it in results. Regarding the use of NOC, in our previous study: “Morales Asencio, J. M., Porcel-Gálvez, Ana María, Oliveros Valenzuela R., Rodríguez Gómez S., Sánchez Extremera L., Barrientos Trigo S. Design and validation of the INICIARE instrument, for the assessment of dependency level in acutely ill hospitalised patients. Journal of Clinical Nursing. 2015. 24(5-6), 761–777”. In this study, we concluded that the inter-observer reliability was always evaluated between two raters and was assessed from the observations performed by clinical nurses who participated in the study. In addition, regarding to the inter-observer reliability, there is good consistency in the individual and global items, as evidenced by the use of the NOC in studies that have evaluated this aspect. We have also included this text under discussion with different bibliographic references that support it.

Furthermore, thank you for your advice, we have included the methodology issues in the limitations of the study.

We hope that the modifications meet your expectations and that you can continue to consider the manuscript for publication in your journal.

Thank you for your time and consideration.

Sincerely, the authors.